The influence of stand composition and season on canopy structure and understory light environment in different subtropical montane Pinus massoniana forests

Jin Peng
Xu Ming mingxu566@163.com
Yang Qiupu
Zhang Jian zhangjian12102@163.com
Key Laboratory of Plant Resource Conservation and Germplasm Innovation in Mountainous Region (Ministry of Education), Collaborative Innovation Center for Mountain Ecology & Agro-Bioengineering (CICMEAB), College of Life Sciences, Guizhou University, Guiyang , Guizhou Province , China
Montagnani Leonardo
Electronic publication date: 2024 Mar 15
Publication date: 2024
Volume: 12
Electronic Location ID: e17067
Received 2023 Oct 3; Accepted 2024 Feb 18
Copyright: © 2024 Jin et al.
Copyright year: 2024
Copyright holder: Jin et al.
License: This is an open access article distributed under the terms of the Creative Commons Attribution License, which permits unrestricted use, distribution, reproduction and adaptation in any medium and for any purpose provided that it is properly attributed. For attribution, the original author(s), title, publication source (PeerJ) and either DOI or URL of the article must be cited.
License URL: https://creativecommons.org/licenses/by/4.0/

Keywords: Understory light environment, Canopy openness, Leaf area index, Pinus massoniana, Mixed forest, Subtropical montane

Funding: National Natural Science Foundation of China 31960234 and 31660150 This work was supported by the National Natural Science Foundation of China (No. 31960234 and 31660150). The funders had no role in study design, data collection and analysis, decision to publish, or preparation of the manuscript.

==============================
Canopy structure and understory light have important effects on forest productivity and the growth and distribution of the understory. However, the effects of stand composition and season on canopy structure and understory light environment (ULE) in the subtropical mountain Pinus massoniana forest system are poorly understood. In this study, the natural secondary P. massoniana—Castanopsis eyrei mixed forest (MF) and P. massoniana plantation forest (PF) were investigated. The study utilized Gap Light Analyzer 2.0 software to process photographs, extracting two key canopy parameters, canopy openness (CO) and leaf area index (LAI). Additionally, data on the transmitted direct (Tdir), diffuse (Tdif), and total (Ttot) radiation in the light environment were obtained. Seasonal variations in canopy structure, the ULE, and spatial heterogeneity were analyzed in the two P. massoniana forest stands. The results showed highly significant (P < 0.01) differences in canopy structure and ULE indices among different P. massoniana forest types and seasons. CO and ULE indices (Tdir, Tdif, and Ttot) were significantly lower in the MF than in the PF, while LAI was notably higher in the MF than in the PF. CO was lower in summer than in winter, and both LAI and ULE indices were markedly higher in summer than in winter. In addition, canopy structure and ULE indices varied significantly among different types of P. massoniana stands. The LAI heterogeneity was lower in the MF than in the PF, and Tdir heterogeneity was higher in summer than in winter. Meanwhile, canopy structure and ULE indices were predominantly influenced by structural factors, with spatial correlations at the 10 m scale. Our results revealed that forest type and season were important factors affecting canopy structure, ULE characteristics, and heterogeneity of P. massoniana forests in subtropical mountains.

Introduction

Light is a key environmental factor that provides energy for plant photosynthesis, thus, driving primary productivity in plants and serving as the primary source of energy input for all forest ecosystems. Light conditions are intimately tied to many ecological processes, compositional structures, and ecosystem service functions in forest ecosystems, such as the diversity of understory plants (Sercu et al., 2017), species composition (Musselman, Pomeroy & Link, 2015), biodiversity (Bartels & Chen, 2010) and global climate regulation (Nakamura et al., 2017). In a given forest ecosystem, only a small amount of light can enter the forest (Jiang et al., 2005) after passing through the forest canopy owing to the combined effects of canopy structure (vertical and horizontal) and dynamic changes (Ediriweera, Singhakumara & Ashton, 2008; Takashima, Kume & Yoshida, 2006). Furthermore, the understory light environment (ULE) exhibits high spatial heterogeneity that provides opportunities for certain tree species to become dominant, thereby enhancing forest productivity. However, most recent studies have focused on the average light levels at the forest floor, while ULE spatial heterogeneity and its underlying mechanisms have been largely overlooked despite their crucial importance for forest sustainability (Feng et al., 2022; Ligot et al., 2016). Therefore, exploring the heterogeneity of canopy structure and ULE in different forest types is important for enhancing forest productivity and sustainable management.

Mixed forests (MFs) have higher productivity than pure forests (Feng et al., 2022; Kelty, 2006), and they can use resources more effectively and stably in response to global climate change (Pretzsch et al., 2015). The “mixing effect” may be due to vertical stratification, spatial complementarity and plasticity of canopies, and niche differentiation among MFs species (Pretzsch, 2014; Sapijanskas et al., 2014). Therefore, each layer of MFs can effectively utilize light resources which are highly correlated with productivity (Erskine, Lamb & Bristow, 2006; Thomas et al., 2023), thereby increasing forest diversity and productivity. Currently, tree species mixing has been widely promoted as a promising silvicultural tool to improve productivity or ecological service functions in artificial forests. For example, forest management in Europe has shifted from a production forestry model dominated by logging to “close-to-nature forest management” (Lu et al., 2018). In Asia, China is transforming artificially pure forests into heterogeneous MFs (Yang, 2022). Some tropical countries emphasize cultivating MFs of native species with high economic value (Erskine, Lamb & Bristow, 2006), integrating local environmental features such as light conditions. Although the area of MFs is increasing, the mechanisms of mixing effects in MFs remain unclear (Liu, Kuchma & Krutovsky, 2018), including suitable mixing methods based on specific site conditions and the biological characteristics of tree species. Therefore, selecting the optimal tree species combinations with low light competition, high economic value, and ecological services based on canopy and light environment mechanisms remains crucial for establishing MFs (Oxbrough et al., 2012).

Seasonal changes (solar elevation angle, temperature) and canopy structure (leaf area index) are important factors affecting the forest light environment. Deciduous tree species have distinct active (growing) and dormant (non-growing) periods. During the growing season, the budding and leaf expansion of canopy leaves increase the light absorption capacity of the canopy, decreasing light intensity beneath the trees. As deciduous canopy leaves fall in the non-growing season, light intensity beneath the forest canopy increases. Therefore, the ULE in forests with deciduous tree species often exhibits pronounced seasonal differences (Zhou et al., 2021). The ULE in evergreen tree forests is generally less affected by combined changes between seasons and canopy structure; the canopy structure of evergreen broadleaved forests does not change significantly seasonally, and the canopy has a high light-intercepting ability. Therefore, evergreen coniferous forests maintain relatively low-light environmental conditions throughout the year (Li & Li, 2003). Additionally, changes in canopy structure across different seasons can increase the spatiotemporal diversity of sun flecks in the understory, thereby increasing the spatial heterogeneity of the ULE (Leakey, Scholes & Press, 2005). This provides opportunities for colonization and growth of different species in the understory, thereby increasing species diversity in forest ecosystems and enhancing ecosystem stability, which is advantageous in mitigating the impacts of climate change.

Subtropical montane forests redistribute environmental factors such as light, temperature, and humidity, affecting the growth of understory plants, impacting forest composition and resource availability (Wang et al., 2023). These environmental changes significantly shape the crown structure and the ULE by influencing plant growth and development. However, little is known about the factors influencing canopy structure and the ULE in subtropical montane forests and their distributional characteristics. Therefore, in this study, we investigated typical natural secondary Pinus massoniana—Castanopsis eyrei MFs and P. massoniana plantation forests (PFs) in the subtropical mountains of southwest China. Our first hypothesis is that the canopy structure and the ULE will differ significantly among the different P. massoniana forest types. Our second hypothesis is that the canopy structure and the ULE is affected by seasonal changes between pure and mixed P. massoniana forest stands. Ultimately, this study aimed to enrich ecological theory and provide a scientific basis for the sustainable management of subtropical montane P. massoniana forests.

Materials and Methods

Study area

The research site is situated in Kaiyang County, Guizhou Province, Southwest China (106°45′–107°17′E, 26°48′–27°22′N). It is characterized by a predominantly mountainous terrain and ranges in elevation 1,000–1,400 m. The region belongs to a mid-subtropical humid monsoon climate, with an average yearly temperature around 15 °C. Its vegetation is primarily evergreen broad-leaved forests, occupying 52.92% of the area, and the forest soil is predominantly yellow or yellow-brown loam. P. massoniana and Cunninghamia lanceolata are primary species of the current artificial forests, whereas Fagaceae, Pinaceae, Theaceae, Aquifoliaceae, and Lauraceae plants account for the majority of species in secondary natural forests.

Site selection and survey

As study sites, we selected a natural secondary forest of P. massoniana–C. eyrei mixed forest (MF) and P. massoniana plantation forest (PF), in each plot of which, we established a single 50 m × 50 m survey quadrat (Table 1). Using the adjacent grid method, each survey quadrat was further divided into one hundred 5 m × 5 m grid cells with the aid of a Real-time kinematic measuring instrument (RTK: Beidou Haida TS5 Pro, Hi-Target, China), and for each cell, we recorded the central latitude and longitude. For each tree in each grid cell with a diameter at breast height (DBH) of greater 5 cm, we recorded the species, DBH, height, and crown width.

Table 1 Basic descriptive information of sites.

Sites	Latitude and longitude	Altitude(m)	Aspect(°)	Slope(°)	height(m)	Mean diameter(cm)	Basal area(cm2)	Stand density (N/hm2)	
P. massoniana plantation forest (PF)	26°57′56″N, 106°54′18″E	1,219	224	12	11.3	17.4	303.1	1,484	
P. massoniana-C. eyrei mixed forest (MF)	26°58′3″N, 106°54′10″E	1,178	241	5	10.0	13.7	207.7	1,356	
Note:

N, Number of measured trees per forest types.

Data collection

During mid-July and late December in 2022, canopy and ULE information was collected using a Top-1300 plant canopy image analyzer (Top-1300, Hangzhou, China). To obtain data, points were placed in the centers of each 5 m × 5 m cell to take photographs (Matsuo, Hiura & Onoda, 2022), with images being obtained from directly above using a 180-degree fisheye lens positioned at a height of 1.3 m above ground level. To avoid interference from direct sunlight, photography was performed during the early morning, at dusk, or on cloudy days.

Calculations and analyses

Photographs were processed using Gap Light Analyzer 2.0 software (https://rem-main.rem.sfu.ca/forestry/publications/downloads/gaplightanalyzer.htm) to obtain two canopy parameters, canopy openness (CO) and leaf area index (LAI), as well as the amount of transmitted direct (Tdir), diffuse (Tdif), and total radiation (Ttot) light environment data (Frazer et al., 2001).

Non-parametric tests were performed on the data using R 4.21 software to analyze the significance between the canopy structure and light environment indicators. The non-metric multidimensional scaling method (NMDS) was used to rank the canopy and ULE of different P. massoniana forest types and different seasons, and then analysis of similarities (ANOSIM) to assess the presence of significant differences between groups. GS+7.0 software was used to calculate the semi-variance function and fit the theoretical model to analyze the spatial heterogeneity of the canopy structure and ULE. Spatial distribution maps of canopy and light environment indicators were constructed using Kriging interpolation in the Geostatistical Analyst module of ArcGIS 10.2 software. The ‘ggplot2’ R package was used for visual mapping.

Results

Differences in canopy structure and ULE in different P. massoniana forest types

Canopy (CO, LAI) and ULE (Tdir, Tdif, Ttot) indicators show extremely significant differences (P < 0.01) among different P. massoniana forest types, as well as significant differences between seasons. The coefficient of variation for all indicators in MF is higher than that in PF (except for LAI in summer), with the Tdir coefficient of variation being the highest. The CO in MF is significantly lower than that in PF for Tdir, Tdif and Ttot, while the LAI in MF is significantly higher than that in PF. The CO in summer is lower than that in winter, and the LAI and ULE indicators in summer are all significantly higher than those in winter (Fig. 1, Table S1).

Figure 1 Differences in canopy structure and understory light environment of PF and MF in summer and winter.

Canopy structure, which includes CO (A) and LAI (B); understory light environment, which includes Tdir (C), Tdif (D) and Ttot (E).

The NMDS indicates significant differences in canopy and ULE indicators among forest types and seasons (Fig. 2). The NMDS stress values for the canopy structure and the ULE are both less than 0.05, and the square of R is close to 1, indicating a good fit (Bray & Curtis, 1957). The results of NMDS combined with ANOSIM show that forest type and season significantly affect canopy structure (Fig. 2A, Stress = 0.012, R = 0.311, P = 0.01) and the ULE (Fig. 2B, Stress = 0.012, R = 0.246, P = 0.01).

Figure 2 Non-metric multidimensional scaling ordination graph of canopy structure (A) and understory light environment (B) in PF and MF at summer and winter seasons.

Canopy structure, which includes CO and LAI (A); understory light environment, which includes Tdir, Tdif and Ttot (B).

Spatial variability characteristics of canopy structure and ULE in different P. massoniana forest types

The best-fitting spatial distribution models of different types of P. massoniana canopy structures and ULE indicators are the Gaussian model and spherical mode. The range of the coefficient of determination for each indicator’s model is 0.602 to 0.932, and the residual sum of squares (RSS) is relatively low (Table 2). These models accurately describe the spatial structural characteristics of the canopy structure and ULE. The nugget value (C0+C) for MF is higher than that for PF, indicating greater fluctuations in the MF canopy structure and ULE. The spatial structure ratio (C/(C0+C)) for the two forest types for each indicator exceeds 90%, suggesting strong spatial autocorrelation. This means that variations caused by spatially structured factors (stand, climate, soil physical and chemical properties) account for 90%, while variations caused by random factors (felling, fertilization and tillage) account for less than 10% (Yan, He & Yang, 2020). The range for different types of P. massoniana canopy structures and ULE indicators is within 10 m, indicating the spatial correlation of indicators within this range.

Table 2 Semivariogram models and parameters for canopy structure and understory light environment.

Sites	Season	Indicator	Model	Nugget
C0	Sill
C0+C	Range
(m)	Spatial structure ratio
C/C0+C	Coefficient of determination
R2	Residual sum of squares
RSS	
PF	Summer	CO	Gau	0.001	0.843	7.985	0.999	0.721	0.032	
LAI	Gau	0.001	0.789	9.405	0.999	0.724	0.057	
Tdir	Gau	0.093	0.965	6.668	0.904	0.932	0.002	
Tdif	Sph	0.001	0.842	8.230	0.999	0.722	0.014	
Ttot	Gau	0.001	0.874	8.833	0.999	0.816	0.031	
Winter	CO	Gau	0.001	0.828	7.742	0.999	0.602	0.046	
LAI	Gau	0.001	0.750	8.972	0.999	0.640	0.066	
Tdir	Sph	0.001	0.966	7.000	0.999	0.617	0.011	
Tdif	Gau	0.020	0.806	7.032	0.975	0.733	0.010	
Ttot	Gau	0.001	0.833	8.158	0.999	0.653	0.050	
MF	Summer	CO	Sph	0.001	0.918	6.350	0.999	0.363	0.008	
LAI	Gau	0.001	1.026	7.898	0.999	0.633	0.073	
Tdir	Gau	0.083	1.096	7.621	0.924	0.879	0.012	
Tdif	Gau	0.001	0.972	7.760	0.999	0.594	0.072	
Ttot	Gau	0.001	1.056	7.604	0.999	0.725	0.039	
Winter	CO	Gau	0.001	1.066	8.678	0.999	0.792	0.047	
LAI	Gau	0.001	1.004	9.405	0.999	0.725	0.081	
Tdir	Sph	0.021	1.012	7.710	0.979	0.927	0.002	
Tdif	Sph	0.001	1.028	9.059	0.999	0.831	0.039	
Ttot	Gau	0.001	0.994	8.245	0.999	0.797	0.030	
Note:

Sph, Spherical model; Gau, Gaussian model.

Canopy spatial distribution characteristics in different P. massoniana forest types

The spatiotemporal distribution of canopy indicators (CO, LAI) shows significant differences among different P. massoniana forest types. The number and distribution of high- and low-LAI patches differ under different forest types; MF has lower heterogeneity than PF. The CO distribution trend in different seasons also differs. PF exhibits an obvious but relatively gentle decreasing trend in summer, while the MF summer trend is severe (Fig. 3). Therefore, canopy heterogeneity is influenced by forest type and season, with PF exhibiting the highest LAI heterogeneity.

Figure 3 Kriging interpolation of canopy structure in different forest types during summer and winter seasons.

Canopy structure, which includes CO (A, B, E, F) and LAI (C, D, G, H).

ULE spatial distribution characteristics in different P. massoniana forest types

The spatial and temporal distribution of ULE indicators varies significantly among different P. massoniana forest types. Among different forest types, the number of Tdir patches is higher in MF than in PF, and the distribution of Ttot is more uniform in MF than in PF. The high and low Tdir values appear in different regions during the summer, and changes are pronounced; however, the Tdir distribution is more uniform, and changes are smoother during the winter (Fig. 4).

Figure 4 Kriging interpolation of understory light environment in different forest types during summer and winter seasons.

Understory light environment, which includes Tdir (A, B, E, F), Tdif (C, D, G, H) and Ttot (I, K, J, L).

Discussion

Differences in canopy structure and the ULE in different P. massoniana forest types

Forest type affects forest canopy structure and ULE characteristics (Niinemets, Cescatti & Christian, 2004). Our results indicated significant differences in canopy structure between two types of P. massoniana stands (P < 0.01). Compared to PF, MF had a lower CO and higher LAI (Figs. 1 and 2, Table S1). In MF, P. massoniana is a pioneer tree species in the forest community succession, with a high and well-developed canopy (horizontal longer crown). In contrast, C. eyrei is a shade-tolerant species in the later stage of succession, with lower canopy height and larger leaf area. Therefore, the two tree species can occupy different spaces in the MF, characterized by low and high CO and LAI, respectively. The LAI of P. massoniana plantations exhibited high heterogeneity (Fig. 3). Firstly, PF is in the mature forest stage (i.e., it has undergone self-thinning), and its stand density is lower and uneven (Wang et al., 2021; Yao et al., 2022). P. massoniana has a denser crown (higher leaf density per unit volume), resulting in it having more gaps that drive the highly heterogeneous nature of the light conditions of P. massoniana forest on the horizontal axis. Secondly, intraspecific competition of P. massoniana led to a reduction in the crown cross-sectional area and taper, as well as an increase in tree height (Del et al., 2019), which increased light heterogeneity on the vertical axis. Therefore, the LAI of P. massoniana has high heterogeneity in PF.

The ULE is related to forest type (Hardy et al., 2004), and canopy tree species composition can significantly affect the light environment within the stand (Montgomery & Chazdon, 2001). In this study, all ULE indicators in different P. massoniana forest types showed significant differences (Figs. 1 and 2), with Tdir exhibiting the highest spatial distribution heterogeneity. P. massoniana, with its small leaf area, clustered leaves, and high tree height, exhibits low sunlight interception efficiency throughout the year (Ruiz et al., 2021). Moreover, it often forms gaps between adjacent trees (Messier, Parent & Bergeron, 1998). In contrast, C. eyrei canopy leaves have a greater capacity for absorbing, reflecting, refracting, and transmitting light. Previous studies have indicated that the crowns of tree species in MFs tend to expand (Jucker, Bouriaud & Coomes, 2015), forming an overlapping, multilayered canopy structure (Lu et al., 2018; Morin et al., 2011; Pretzsch, 2014) that can intercept light more efficiently. However, light can penetrate directly to the forest floor through forest gaps. Moreover, physical leaf and branch damage and seasonal variations in the angle of solar elevation can contribute to canopy heterogeneity (Lemos-Filho, Barros & Dantas, 2010; Niinemets, Cescatti & Christian, 2004; Purves, Lichstein & Pacala, 2007). These changes in canopy structure significantly impact the ULE, ultimately leading to heterogeneity in understory light conditions (Clinton, 2003; Valladares & Guzmán, 2006). However, Tdif comes from different directions in the sky (Čater, Schmid & Kazda, 2012), and through the absorption and reflection of different canopy heights and thicknesses, it tends to be uniform once on the forest floor. In contrast, Tdir is highly transmissible, allowing it to directly reach the forest floor and exhibit high heterogeneity. Overall, unlike Tdir, Tdif exhibits little spatiotemporal variation (Promis et al., 2009).

Seasonal effects on canopy structure and the ULE in different P. massoniana forest types

Seasonal changes, which lead to variations in climatic factors like light, temperature, and rainfall, can directly affect plant physiological processes (shedding of branches and leaves (Zhang et al., 2014)), which in turn can affect canopy structure (Tang & Dubayah, 2017). We found significant differences among canopy structure indices across seasons (Fig. 1, Table S1); these exhibited distinct spatial distributions within different types of P. massoniana stands, with summer having a lower CO and higher LAI than winter (Fig. 3). Different tree species exhibit seasonal variations due to their inherent biological characteristics. In this study, 2-year-old P. massoniana needles showed phenological changes with autumn shedding and spring budding (Zhang et al., 2014); in contrast, the deciduous periods for evergreen Fagaceae species like C. eyrei mainly occur in April and November (Li et al., 2014), leading to certain seasonal changes in canopy structure (Chen, 1996). Additionally, there were other deciduous tree species (e.g., Choerospondias axillaris, Liquidambar formosana, etc.) in the MF habitat, and the seasonal changes can also shape canopy structure. Season is closely related to leaf angle (Raabe et al., 2015), which affects leaf distribution and arrangement, which, in turn, has a seasonal effect on canopy structure. All these factors result in canopy structure, seasonal differences, and spatiotemporal heterogeneity.

Seasonal dynamics and related changes are equally important for the characteristics and distribution patterns of the ULE (Tang & Dubayah, 2017). Seasonal canopy dynamics are closely related to sunlight (Zhao et al., 2022). When the canopy is influenced by leaf phenology (bud burst and leaf expansion in spring and summer, and leaf drop in autumn and winter), solar elevation angle (intense and longer sunlight in summer, weaker and shorter sunlight in winter, (Lemos-Filho, Barros & Dantas, 2010)), and cloud cover (Weiss, 2000), the forest ULE changes accordingly (Hartikainen et al., 2020). Therefore, ULE indices in different seasons all displayed significant differences and spatiotemporal heterogeneity (Figs. 2 and 4, Table S1). In addition, lower P. massoniana light interception efficiency increases the survival possibilities of other broadleaf tree species in the understory (Ruiz et al., 2021), leading to trees having canopies of varying heights in different seasons, which can effectively absorb and reflect light. As a result, the Tdif fluctuations in the ULE across different seasons were minimal, and its distribution was relatively uniform (Promis et al., 2009).

Similar to a majority of the studies of this type, the design of the present study has certain limitations. Firstly, there was no repetition of sample plots, and the field test was potentially influenced by multiple factors, not all of which could be controlled. The study area is located in a typical karst area of southern China, the terrain of which is rugged and changeable. As far as possible, however, to ensure valid comparisons between plots, we attempted to select sites with comparable features (similar horizontal distance, altitude, and slope direction). Secondly, with respect to photography, when distinguishing between canopy and non-canopy, it is necessary to set an artificial threshold, which typically involves a certain element of subjectivity. In order to resolve this problem, all photographs were processed by the same person, thereby minimizing the likelihood of assessor error. To gain more robust data, further long-term observations at different spatial scales (different latitude zones, different vertical heights of forests, and shrub and grass shading) can provide additional insights into the adaptation of canopy structure and light to climate change over extended periods of time, thereby enabling us to gain a more comprehensive understanding of forest interactions.

Conclusions

We found significant differences (P < 0.01) in both canopy structure and the ULE between the two types of P. massoniana forest stands. These statistically significant differences were also observed across different seasons. Overall, these results showed that P. massoniana forest composed of different tree species will exhibit differences in spatiotemporal variability, resulting in differences in light conditions. For example, P. massoniana MF composed of more tree species has more complex canopy structure and higher light utilization rate. This study emphasizes the need for managers to consider the diversity of canopy species and heterogeneity of light environments when converting mixed P. massoniana plantation forests in subtropical montane regions.

Supplemental Information

Supplemental Information 1 Descriptive statistics of Canopy structure and understory light environment.

Supplemental Information 2 Raw data of Figures 1 and 2.

Supplemental Information 3 Raw data of Figure 3.

Supplemental Information 4 Raw data of Figure 4.

Supplemental Information 5 Raw data of Table 1.

We wish to thank the editor and anonymous reviewers for their constructive comments and suggestions for improving this manuscript.

Additional Information and Declarations

Competing Interests

Author Contributions

Data Availability

The authors declare that they have no competing interests.

Peng Jin performed the experiments, analyzed the data, prepared figures and/or tables, authored or reviewed drafts of the article, and approved the final draft.

Ming Xu conceived and designed the experiments, performed the experiments, analyzed the data, authored or reviewed drafts of the article, and approved the final draft.

Qiupu Yang performed the experiments, analyzed the data, authored or reviewed drafts of the article, and approved the final draft.

Jian Zhang conceived and designed the experiments, performed the experiments, analyzed the data, prepared figures and/or tables, authored or reviewed drafts of the article, and approved the final draft.

The following information was supplied regarding data availability:

The raw measurements are available in the Supplemental Files.

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
