# Peer review of "The influence of stand composition and season on canopy structure and understory light environment in different subtropical montane Pinus massoniana forests"

_PeerJ, doi:10.7717/peerj.17067_

## Round 0.1 · original submission · Major Revisions

Dear Ming Xu, dear Jian Zhang,

We received two evaluations of your paper and I also have considered it. One reviewer was cursory, making a simple comment, while the second was more detailed and in my view gave helpful indications.

My comments are in line with the second reviewer. When presenting a study related to forests is always helpful to present basic forest information, like tree dominant height, basal area, age, and presence or absence of tree saplings. Are you able to assess the temporal variation in LAI and light in the understory? What is the frequency distribution (not only the spatial distribution) of light intensity at the ground? Could you define the Gini coefficient of the heterogeneity of the two forest types? How was the thinning done in the two forest types? Any indication about the vertical profile of leaf area density?
Finally, I disagree with the conclusion indicating that the forests displayed seasonal structural differences, simply because you found different light conditions. Light conditions change during the season also since the solar angle is changing.

Mine are simply suggestions, with the hope that the study can become ecologically relevant.

Sincerely,

Leonardo Montagnani

Reviewer 1 ·

Basic reporting

I really liked the article. This is due to the fact that the relevance is not in doubt, especially due to the detailed experimental and parameters.

Experimental design

no comment

Validity of the findings

In addition, the need to restore forests after anthropogenic and natural impacts is becoming an increasingly significant problem. Forest ecosystems have a complex structure, so it is necessary to have a clear understanding of the ways of their development when removing anthropogenic impact.

Additional comments

To facilitate transparent and open science, we encourage authors to publish their results and experimental methodology in as much detail as possible so that results can be reproduced

Reviewer 2 ·

Basic reporting

Crown structure and understory light conditions have a significant impact on forest productivity and related ecological processes. The author studied the crown structure and light environment parameters of natural secondary mixed forests (MF) and artificial forests (PF) of Masson pine and found that forest type and season had a significant influence on the crown structure and ULE index of Masson pine forests. Forest type and season are important factors affecting the crown structure, understory light characteristics, and heterogeneity of subtropical mountainous Masson pine forests. This research contributes to the sustainable management of subtropical Masson pine forests.

Experimental design

Crown structure is related to forest composition and age. Natural forests are usually composed of different-aged stands, resulting in more complex forest structures. It may be better to compare two artificial forests with the same age but different compositions.

Validity of the findings

The conclusions drawn are also obvious (there are differences in the crown structure and light environment between natural forests and pure forests; seasons may affect the angle of light, and the change in forest canopy may be caused by deciduous tree species, resulting in differences in light environment between seasons). The conclusions need to be refined and summarized based on crown structure (the impact of tree species composition)

Additional comments

1. The article should focus on the research content and eliminate the parts related to forest management.
2. The summary should contribute to posing scientific questions and have appropriate logical relationships. Hypotheses should be stated in declarative and conclusive language, specifically addressing the scientific questions.
3. Basic information such as slope and aspect of the experimental sites, canopy closure, density, tree species composition, tree height, and diameter at breast height of the two forest types, as well as age, understory shrub type, and coverage of artificial Masson pine forests, should be provided as they are closely related to crown structure.
4.Information about the analysis software and instruments used should be added.
Figure 2 legend suggestion: use different legends for natural forest and artificial forest, with the summer being blank and the winter being filled.
5. Specific writing suggestions:
- L37-41: The sentence is confusing. I would consider deleting the parentheses and using shorter sentences.
- L37: change to "ecosystems"
- L40: (Bartels & Chen, 2010)
- L53: use "in response" instead of "when responding"
- L57: use "which" instead of "that"
- L73: add "the" before "budding"
- L87-89: I don't understand what this sentence is trying to convey. It is suggested to rewrite it as shorter sentences.
- L102: Other information about the research site should be added, such as stand age and topography.
- L143: Should "P" be capitalized?
- L191-192: L196-201: There is too much explanation in parentheses, which makes the content messy.
- L218-221: The relationship between these two sentences and the issue being explained is not clear.
- L225: Change "thus impacting" to "which in turn can affect" may be better.
- L233: Additionally, there is a problem with the sentence structure.
- L246: Additionally should be changed to In addition,
- L253: "Firstly, there was no repetition of sample plots, and the field test was potentially influenced by multiple factors, not all of which could be controlled" undermines the reliability and credibility of this study.
- L269: Move Furthermore to the next sentence.
- L270: "They indicated that" is not the correct wording. It should be removed.

Annotated reviews are not available for download in order to protect the identity of reviewers who chose to remain anonymous.

---

## Round 0.2 · Minor Revisions

Dear Ming Xu, dear Jian Zhang,

We received two evaluations of your study.

Both reviewers are satisfied with the improvements done.

Nevertheless, I noticed a few points that could be improved.

1) Can you specify, in the figure captions, the content of all the panels reported in the composite figures?

2) Can you use a less subjective term than 'better canopy structure' in the conclusions section? A simplified structure can be preferable for harvesting purposes.

Sincerely,

Leonardo Montagnani

Reviewer 1 ·

Basic reporting

no comment

Experimental design

no comment

Validity of the findings

no comment

Additional comments

After the paper was revised, the language became more fluent, some ambiguities were corrected, and the charts became clearer

Reviewer 2 ·

Basic reporting

No comment

Experimental design

No comment

Validity of the findings

No comment

Additional comments

No comment

---

## Round 0.3 · accepted · Accept

Dear Dr. Jin and Dr. Zhang,

I am pleased to inform you that I consider your paper acceptable now.

Sincerely,

Leonardo Montagnani